# The Exploitation of the Glycosylation Pattern in Asthma: How We Alter Ancestral Pathways to Develop New Treatments

**DOI:** 10.3390/biom14050513

**Published:** 2024-04-24

**Authors:** Angelika Muchowicz, Agnieszka Bartoszewicz, Zbigniew Zaslona

**Affiliations:** Molecure S.A., Zwirki i Wigury 101, 02-089 Warszawa, Poland; a.muchowicz@molecure.com (A.M.); a.bartoszewicz@molecure.com (A.B.)

**Keywords:** asthma, glycosylation, glycobiology, biomarkers, chitotriosidase 1

## Abstract

Asthma has reached epidemic levels, yet progress in developing specific therapies is slow. One of the main reasons for this is the fact that asthma is an umbrella term for various distinct subsets. Due to its high heterogeneity, it is difficult to establish biomarkers for each subset of asthma and to propose endotype-specific treatments. This review focuses on protein glycosylation as a process activated in asthma and ways to utilize it to develop novel biomarkers and treatments. We discuss known and relevant glycoproteins whose functions control disease development. The key role of glycoproteins in processes integral to asthma, such as inflammation, tissue remodeling, and repair, justifies our interest and research in the field of glycobiology. Altering the glycosylation states of proteins contributing to asthma can change the pathological processes that we previously failed to inhibit. Special emphasis is placed on chitotriosidase 1 (CHIT1), an enzyme capable of modifying LacNAc- and LacdiNAc-containing glycans. The expression and activity of CHIT1 are induced in human diseased lungs, and its pathological role has been demonstrated by both genetic and pharmacological approaches. We propose that studying the glycosylation pattern and enzymes involved in glycosylation in asthma can help in patient stratification and in developing personalized treatment.

## 1. Introduction

Asthma is a complex condition that is widely acknowledged as a heterogeneous disorder, previously classified into “allergic” and “intrinsic” subtypes [1]. Based on the significant advancements in understanding asthma, this two-type categorization is an oversimplification [2]. Still, the mechanism of the disease can be described as T-helper-2-associated asthma (Th-2 form) or no Th-2 type. Th-2-associated asthma usually develops during childhood and can be prompted by early contact with common environmental allergens, for example, animal dander or pollen [3]. Allergen presentation leads to T cell activation and polarization into the Th-2 type, which produces IL-4, IL-5, and IL-13. These cytokines and T cells further support IgE-allergen-specific antibody production by B cells and the influx and accumulation of eosinophils in the airway wall [4]. In contrast, non-allergic asthma typically develops in adulthood and is often linked to factors such as obesity, aging, and smoking. This type of asthma seems to involve the dysregulation of multiple pathways and can be difficult to treat [5]. Based on symptoms, asthma can be classified as mild, moderate, or severe, each of which requires specific treatment [6]. When intensive treatment fails to relieve symptoms and poses a life-threatening risk, then the patients deal with severe asthma, which can develop in both children and adults. Therefore, treatments that consider cellular and molecular mechanisms are necessary to control symptoms and reduce attacks, especially in severe cases of asthma.

The primary methods of asthma diagnosis still involve patient questionnaires, allergy testing, and lung-function-based criteria, which are typically sufficient for most patients who respond positively to modern standard therapy [7]. The conventional way of describing disease makes asthma a good case for studies developing molecular patterns of each subset, which can personalize treatments. In cases of severe asthma, a patient’s condition may not improve, even with intensive treatment, which often results in death [8,9]. Despite attempts to subtype asthma, its complexity makes classification difficult, carrying the risk of inappropriate therapy, especially for patients with severe symptoms [10]. Therefore, it is crucial to find additional diagnostic markers that reflect the complex mechanism of this disease [11]. Discovering biomarkers would also provide valuable information about the individual mechanism of the asthma stage and the main cellular players of each patient, enabling the development of a personalized approach to managing and treating their asthma effectively. Although protein glycosylation patterns have been used as biomarkers in diseases such as diabetes [12,13], they have not been extensively utilized in asthma. This review aims to outline a strong rationale for studying glycosylation in asthma to better understand the disease mechanism and help assign it to the appropriate subtype and treatment regime.

## 2. Protein Glycosylation

### 2.1. Definition of Protein Glycosylation

Protein glycosylation, one of the most ancient and complex post-translational modifications, encompasses N-linked glycosylation and O-linked glycosylation, collectively affecting over 50% of known proteins [14]. It plays critical roles in various biological processes, including protein folding, stability, trafficking, recognition, cell adhesion, ligand binding, and signaling [15]. Moreover, it can modulate protein–protein interactions and influence the immunogenicity of proteins. The dysregulation of protein glycosylation has been implicated in various diseases [16,17], including asthma [14]. This modification is crucial for the structure, stability, and function of many crucial proteins in living organisms. N-linked glycosylation relates to glycans that are covalently linked to the protein via the nitrogen atom of the asparagine residues within the consensus sequence Asn-X-Ser/Thr (where X can be any amino acid except proline) [18], while O-linked glycosylation involves the oxygen atom of the serine or threonine. Unlike N-linked glycosylation, O-linked glycosylation does not require a consensus sequence, rendering it more variable and less predictable. O-GlcNAcylation is a dynamic modification that involves the reversible attachment of a monosaccharide to serine or threonine residues of proteins. The majority of the process of protein glycosylation occurs in the endoplasmic reticulum and Golgi apparatus, as well as the cytoplasm. This is due to the availability of substrates and the presence of appropriate enzymes. Protein glycosylation is influenced by a number of regulatory mechanisms available [19] as well as the overall energetic state of a cell as it affects the availability of metabolic precursors required for this process [20].

Protein glycosylation is regulated by metabolic flux and influenced by altered glycolysis, which is commonly implicated in immune cell activation and proinflammatory functions [21]. The hexosamine biosynthetic pathway (HBP), responsible for synthesizing substrates utilized in N-glycosylation, plays a significant role in regulating glycolysis. This regulation occurs because both pathways share the initial steps of converting glucose to glucose-6-phosphate and fructose-6-phosphate. As glycolysis rapidly converts glucose to produce adenosine triphosphate (ATP), glycosylation, facilitated by the HBP pathway, is closely linked to the energy status of the cell [22,23].

### 2.2. Building Blocks of Glycosylation

The synthesis of glycans in humans is a dynamic and highly regulated process that involves the coordinated action of numerous enzymes and substrates within various cellular compartments [24,25]. The dysregulation of glycan synthesis can lead to various diseases and disorders, highlighting the importance of understanding the mechanisms underlying glycan biosynthesis. Glycans are synthesized through both enzymatic and non-enzymatic pathways, and they primarily occur in the endoplasmic reticulum (ER) and Golgi apparatus of cells [26,27].

The building blocks for glycan synthesis are activated nucleotide sugars, which are synthesized in the cytoplasm of the cell. These nucleotide sugars serve as donors of specific sugar residues for glycan synthesis [28]. For example, UDP-glucose, UDP-galactose, GDP-mannose, and UDP-N-acetylglucosamine are common nucleotide sugars used in glycan biosynthesis. Further glycan synthesis from oligosaccharides typically begins in the ER, where the initial steps involve the transfer of a preassembled oligosaccharide precursor (usually a lipid-linked oligosaccharide) onto specific asparagine residues of nascent polypeptide chains. This process, known as N-linked glycosylation, involves the action of glycosyltransferases and oligosaccharyltransferase enzymes [29]. Following the initial glycan attachment, further processing and elongation of the glycan chains occur as the proteins move through the Golgi apparatus. Various glycosyltransferases and glycosidases located in different compartments of the Golgi catalyze the addition and removal of sugar residues, leading to the generation of diverse glycan structures.

It is interesting to note that enzymes with primary roles that are not directly connected to N-glycan modification and localized outside the ER can also introduce changes in protein glycosylation. Glycan substrates are commonly found alone or as part of glycoproteins in association with the surface or cytosol of cells that are critical for lung function, namely epithelial cells and macrophages. In particular, LacNAc (N-acetyllactosamine) and LacdiNAc are common glycan structures present in glycoproteins specific for asthma, such as mucin and immunoglobulins (IgG and IgA) [30].

### 2.3. Pathways Driving Protein Glycosylation

The glycosylation process involves different types of enzymes, including two major groups, namely glycosyltransferases, which play a role in the biosynthesis of glycans, and glycosidases, which hydrolyze glycosidic bonds and remove monosaccharide units during the maturation of glycans. Furthermore, glycans can undergo additional modification of sugar units via a range of several enzymes, such as sulfotransferases, phosphotransferases, O-acetyltransferases, O-methyltransferases, pyruvyl transferases, and phosphoethanolamine transferases [31]. These huge groups of enzymes, encoded by around 1% of mammalian genes, have previously been described comprehensively [32,33].

The number of enzymes involved in glycosylation is still growing as the new functions of well-described proteins are being discovered. Chitotriosidase (CHIT-1) activity and expression are elevated and implicated in various inflammatory disorders, including lung diseases, and recently, they have also been linked with glycosylation [34,35]. CHIT1 contributes to macrophage polarization [36], a process detrimentally linked to asthma [37,38,39,40], as well as a complex pathway driving airway remodeling [41,42,43,44]. Another interesting enzyme regulating the process of glycosylation is fucosyltransferase 2 (H blood group), or FUT2 [45]. Fucosylation was shown to be associated with greater airway disease severity, and a knockout of FUT2 significantly reduced lung epithelial fucosylation, attenuated eosinophilic inflammation, and decreased airway hyperresponsiveness in HDM-induced asthma models [46]. Notably, FUTs are also expressed by proinflammatory macrophages and neutrophils, and the expression of FUTs increases upon myeloid cell activation and correlates with the level of proinflammatory cytokines, including TNF [47]. In addition to fucosylation, the modification of glycoproteins by sialidases can also affect the accumulation of immune cells during asthmatic lung inflammation [14].

### 2.4. Glycosylation in Asthma

Chronic inflammation leading to airway remodeling is the main feature of many autoimmune and metabolic disorders, including asthma. Recent studies have described alterations of glycosylation in many pathophysiological conditions; however, mechanistic insights are still lacking [16]. Changes in protein glycosylation contribute to disease pathogenesis through the modulation of biological processes, which we detail in the sections below (Figure 1). Many of them include molecular pathways and fine-tuning inflammatory responses that are integral to the development of asthma [48]. Therefore, we discuss specific subsets of immune cells affected by an altered glycosylation status and explain how these changes influence their functions in pathological processes.

The glycolysation state illustrates the condition of cells infiltrating the lungs affected by asthma [48]. In fact, more research in the field of glycobiology should allow us to track immune cell migration and determine their origin. Altered glycosylation can significantly modify leukocyte circulation and cell metabolism, initiate a shift toward proinflammatory functions, and trigger the secretion of proinflammatory immunoglobulins, ultimately leading to the development of various chronic inflammatory diseases [14]. The dysregulation of glycosylation, hypersialylation, and hyperfucosylationas is implicated as the glycosignature of inflammation is associated with the altered activity of key enzymes in metabolic pathways [49]. In fact, the availability of glycans and sugar precursors and the activity of enzymes that participate in glycosylation determine the metabolic pathways that regulate the primary functions of immune cells.

Modifications of glycosylation observed in asthma are diverse and strongly depend on a particular subset of immune cells. Based on the phenotype and mechanisms involved in asthma, recent publications indicate a new classification in endotypes: type 2-low, type 2-high, and type 2-ultra-high asthma [2]. Type 2-ultra-high asthma, when compared to type 2-high asthma, has been described as a very severe, eosinophilic type and is resistant to corticosteroid treatment [2]. Type 2-high and type 2-ultra-high asthma again involve IL-4, IL-5, and IL-13, while type 2-low asthma, due to the lack of biomarkers, is described as a type with different mechanisms of disease, including IL-1β, IL-6, or neutrophil infiltration [50]. Neutrophils develop extracellular traps as well as granules containing myeloperoxidases and elastase that are cytotoxic for epithelial cells and contribute to inflammation and damage to lung tissue [51]. Non/low-type 2 asthma is associated with a poor response to corticosteroids. The term “severe asthma” is not precisely defined and can vary between different sources [9,52]. Nonetheless, it generally refers to patients who have limited responses to treatment and usually fall into the high/ultra-high and other/low type 2 late-onset asthma groups. The course of the disease is therefore related to the type of cells activated and infiltrating the lung tissue—which are processes regulated by glycosylation. Glycosylation changes can precede immune infiltration; in fact, human epithelial cell culture experiments have shown that glycan changes can arise in the absence of immune cells [48]. The initial pathologic response to asthma-inducing irritants occurs at the interface of major airways and is characterized by changes in cellular glycosylation that drive immune infiltration to the bronchi [48]. One of the key interactions involved in the immune cell influx is the binding of leukocytes sialyl Lewis X (sLeX) glycans with selectins presented on the endothelial cells. The inhibition of this binding by an anti-sLeX monoclonal antibody significantly impairs eosinophil circulation and exerts therapeutic effects in a murine model of allergen-induced asthma [53]. Moreover, the percentage of sLeX and CCR4^+^ memory Th lymphocytes are elevated in the blood of patients with asthma, and the number of 6-sulfoLe^X^-positive Th lymphocytes correlate with the eosinophil count and IgE level [54]. These studies suggest that sialyl glycans have a significant role in the pathogenesis of asthma.

## 3. Key Glycoproteins in Asthma

Tracking glycosylation changes can help us understand the progression of inflammatory diseases, including asthma (Figure 1). One of the pathognomonic features of asthma is the hyperproduction of mucus, which is built with glycoproteins [55,56]. Moreover, asthmatic mucus is built by mucins, among which MUC5AC (Table 1) is also heavily fucosylated by FUT2 [57,58]. It was reported that MUC5AC is overproduced in asthma, while MUC5B is related to lung homeostasis and defense [59,60]. When the production of MUC5AC exceeds that of MUC5B, then it is usually associated with pathologic airway measures such as small airway abnormalities, airway obstruction, and increased exacerbation frequencies [59,61,62,63]. Indeed, the widely reported asthma-associated changes in epithelial glycosylation concerns fucosylation, specific alterations in terminal β-linked galactose, N-glycan branching, total GlcNAc, sulfated galactose, and poly-N-acetyl-lactosamine (poly-LacNAc). However, the exact mechanism of glycan inductions has not yet been clearly determined.

Glycans are present on the surfaces of the key entities responsible for inflammatory response, such as adhesion molecules, secreted immunoglobulins, and proteins of the acute phase [16]. Moreover, proinflammatory cytokines, by regulating the expression of glycosyltransferases and the substrate availability required for glycan biosynthesis, can change the composition of glycans [75]. The hallmark of the induction of an allergic lung inflammation event is the activation of the IL13-STAT6 axis, which is responsible for the induction of the expression of FUT2 [48,76,77]. In particular, FUT2 was originally described as an enzyme that controls the ability to secrete ABO blood group antigens in body fluids [78]. The functional nonsense mutation of FUT2 was revealed as the main determinant of secretor/non-secretor status for ABO antigens [79]. The ABO antigens are expressed on various mucosal surfaces, such as the bronchial epithelium, oral mucosa, and gastrointestinal tract, and when secreted, they increase the risk of early childhood asthma [79].

Fucosylation is a form of glycosylation which is well associated with asthma [48]. In a house dust mite (HDM)-induced asthma model the severity of diseases correlated with increased α1,2-linked fucose residues, while a knockout of Fut2 exhibited reduced asthma features [76]. The translational confirmation of the relevance of fucosylation has been described on a sulfated sialyl-LewisX antigen (Sulfo-sLeX), which includes Fuc-α1,3-GlcNAc and a sulfate on the Gal and/or GlcNAc, and has been reported to be increased on peribronchial venules and capillaries [80]. FUT2 is also implicated in C3a complement accumulation, modulating the influx of C3a anaphylatoxin receptor-expressing monocytic-derived dendritic cells (Mo-DCs) in the lungs [76]. Since MO-DCs may present allergens to Th-2 lymphocytes [81], promote T cell stimulation via the OX40 ligand [82], and produce proinflammatory chemokines such as CCL2 and CCL24 (eotaxin 2), the role of FUT2 in MO-DC accumulation is significant in asthma [83].

Mo-DCs are antigen-presenting cells with the ability to take up antigens in the periphery and expose them to T cells to drive lung inflammation [81]. The surfaces of Mo-DCs are covered with glycoproteins decorated predominantly with sialylated glycans, which are regulated during their differentiation and affect Mo-DC functions [84]. DC maturation in the presence of proinflammatory stimuli results in a significant downregulation of the expression and activity of ST6GAL1 and ST3GAL4, which may cause a phenotype switch to inflammatory DCs [85,86]. Interestingly, the role of ST6GAL1 is very relevant in asthma since it plays a critical role in the sialylation of MUC4β in epithelial dysfunction associated with T2-high asthma [66]. Therefore, ST6GAL1 represents a potential target of a specific sialylation pathway in asthma. In fact, the increased expression of both ST6GAL1 and ST3GAL4 was associated with increased goblet and basal-activated cell factors underlying asthmatic susceptibility [87]. While the presence of sialic acids has a tolerogenic effect on DCs, fully desialylated DCs exhibit increased MHC expression, secretion of pro-inflammatory cytokines, phagocytosis, and activation of T cells [88]. Endogenous sialidases, such as neuraminidase 1 and 3 (NEU1 and NEU3), contribute to the desialylation of cells [89,90,91]. In summary, glycosylation pathways controlling both the activation and attenuation of the inflammatory phenotype of DCs have been identified.

It was proposed that modification in cellular glycosylation attracts activated leukocytes to immunogen exposure in the bronchi [92]. Specifically, NEU1 catalyzes the removal of sialic acid and plays an important role in the regulation of signaling in immune cells, including activated T cells [93]. It has been demonstrated that an abnormally high expression of Neu1 correlates with increased immune responses in human respiratory diseases [94,95,96]. NEU1 upregulates the activity of CD44 by increasing hyaluronic acid binding in CD4^+^ lymphocytes [72]. CD44 is a cell adhesion molecule that participates in lymphocyte rolling over inflamed endothelium; specifically, the process of cellular infiltration into asthmatic lungs is critical for the disease’s flares. Moreover, it was shown that not only sialidases, but also the general alternation of the glycosylation of CD44 significantly increases the recognition of its ligands [71,97]. Since adhesion molecules are one of the main proteins implicated in leukocyte trafficking, the glycolytic modification of CD44 impacts Th-2 cell migration and significantly implicates the pathogenesis of acute asthma [48,70,98,99].

T cells, specifically Th-2 cells, play a crucial role in asthma [100]. N-glycosylation of T cells, which significantly affects their functions, is controlled by galectins [101,102]. Gal-1 and Gal-3 preferentially bind to branched N-glycans containing the LacNAc motif found on CD7, CD45, CD43, and TCR, which leads to the inhibited migration and apoptosis of T cells [103]. Beta-1,6-N-acetylglucosaminyltransferase V (MGAT5) is an enzyme that catalyzes the biosynthesis of N-linked glycans and ligands for galectins [104,105]. MGAT5 expression in T cells is altered in many chronic inflammatory diseases, and MGAT5-modified N-glycans have been shown to have a key role in the regulation of allergic airway inflammation [106]. In addition, the IL-10-induced expression of MGAT5 in CD8^+^ T cells promotes the establishment of persistent chronic inflammation [107].

CD25, the subunit of IL2R, is a T cell receptor that is relevant in asthma and a therapeutic target which is heavily N- and O-glycosylated [108,109]. Altered N-glycan branching decreases the surface expression of CD25, which results in the inhibition of IL-2 binding [110,111]. N-glycan branching is not the only feature that influences T cell functions. Gal-1 preferentially kills proinflammatory Th-1 cells over anti-inflammatory Th-2 and Treg cells. The latter is explained by the fact that Th-2 and Treg cells have a higher expression of ST6GAL1, which is responsible for the synthesis of terminal α2,6-sialic acids, compared with Th1 cells, and are thus protected from galectin-mediated apoptosis [112].

An important glycosylation trait of T cells that is altered in chronic inflammation is fucosylation, which plays a crucial role in the migration of T cells to the site of inflammation by facilitating and regulating interactions of selectins and their ligands [113]. The TCR receptor—the one that revolutionized immunotherapy, brought checkpoint inhibitors and CAR T cells to the clinic, and it requires core-fucosylated N-glycans for its proper activation and function [114]. TCR activation is mediated by the Alpha-1,6-fucosyltransferase FUT8, and it is crucial for T cell differentiation into different linages, which shape the pathology of asthma [115]. Core fucosylation is required for the expression of programmed cell death receptor 1 (PD-1), which attenuates TCR signaling [116,117]. PD-1 function is crucial for maintaining T cell homeostasis in allergic diseases, including asthma [118,119], where its expression is regulated, presenting a therapeutic option [120,121].

B cells are yet another important cell type in asthma—specifically due to their secretion of immunoglobulins (Igs), which are the major executive glycoproteins of the humoral adaptive immune response. All human Ig classes are N-glycosylated, with N-glycans affecting the structural stability and conformation of Igs as well as their effector functions. The N-glycosylation of IgG affects its structure and function, causing a potential risk of developing asthma [122]. Moreover, a strong association was reported between increased offspring airway inflammation and pro-inflammatory IgG glycosylation patterns in mothers and offspring [123]. Authors propose that the IgG glycosylation status is an important parameter that should be included in future clinical studies [123].

Another important antibody type in asthma is IgE, which is best known for its role in allergic immune responses. Specifically, IgE binds to high-affinity IgE receptors (FcϵRI) expressed on the surfaces of basophils and mast cells—effector cells of the allergic reaction—triggering release of histamine, lipid mediators, and pro-inflammatory cytokines [124]. IgE is the most glycosylated immunoglobulin, as it has seven N-glycosylation sites [125]. Interestingly there is a single N-glycosylation site at Asn394, which is critical for the IgE-mediated initiation of the allergic cascade, and the complete deglycosylation of Asn394 can actually alter the secondary IgE structure, abolishing FcϵRI binding and the subsequent IgE-mediated allergic reaction [126,127]. This specific site controlling the allergic inflammatory cascade represents an attractive therapeutic opportunity. The binding of another heavily glycosylated antibody, IgA, to FcϵRI can inhibit IgE-mediated asthma [128]. Moreover, in the serum collected from severe patients with asthma, lower levels of IgA were observed in comparison to the healthy controls [129,130]. The whole IgE glycosylation process is endogenously controlled by galectins, namely Gal-3 [131] and Gal-9 [74]. Gal-3 can cross-link IgE and FcϵRI via their N-glycans and trigger allergic reactions, while Gal-9 reduces them by blocking the formation of the IgE-antigen complex [74,131]. Another method of glycoregulating IgE functions is the removal of terminal sialic acid on IgE N-glycans, which attenuate the degranulation of effector cells and subsequent allergic reactions [132].

Severe steroid-resistant asthma is mostly defined by neutrophil presence and neutrophilic marker profiles. Previous trials of antineutrophilic agents have failed because enrolled patients were not specifically chosen for these targeted treatments [133]. This happened because of a lack of proper biomarkers for Th-2 and a low population subset. N-glycosylation has been shown to contribute to important effector functions of neutrophils, such as the extravasation, phagocytosis, degranulation, and formation of neutrophil extracellular traps (NETs) [134]. Neutrophils express the N-glycosylated MAC-1 integrin, which regulates their trafficking, phagocytosis, and interaction with other cells [134]. The neutrophil MAC-1 surface has decreased sialylation and increased the Lewis x structure [Lex, Galbeta1-4(Fucalpha1-3)GlcNAc] (the Lex motif) and high mannose content in chronic inflammation [135]. The Lex motif expressed on MAC-1 mediates binding to DC-SIGN expressed on DCs, thus providing an indirect link between innate and adaptive immunity. Other myeloid cells, such as monocytes and macrophages, also express MAC-1, but since they lack the Lex motif, this trait is exclusively dependent on neutrophils [136]. This is a great example of how carbohydrate decoration offers specificity for therapeutic interventions aimed at preventing lung damage mediated by dysregulated neutrophil trafficking.

### Chitotriosidase—An Emerging Glycoenzyme Linked to the Pathology of Asthma and Interstitial Lung Diseases

Despite the lack of endogenous chitin synthesis, mammalian genomes encode CHIT1, which is a biomarker of various diseases including asthma [137]. Original functions and conventional thinking about chitinase ascribed the role of CHIT1 to an ancient host defense against chitin-containing pathogens, directly promoting inflammation and modulating tissue remodeling and fibrosis [138]. CHIT1 is an ancient enzyme found in humans that belongs to the family of chitinases. In humans, CHIT1 is primarily produced by activated macrophages [139]. It is one of the most abundant chitinase enzymes found in humans and plays a role in immune defense mechanisms. Elevated levels of chitotriosidase in blood serum or other bodily fluids can be indicative of various conditions associated with increased macrophage activity, including asthma. Here, we discuss in detail all of the endogenous substrates for CHIT1, including glycans, in the context of asthma, which we believe far better reflect physiologically relevant substrates for CHIT1 in diseases.

Larsen [34] reported that mammalian-like glycans, namely GlcNAc-containing disaccharides and oligosaccharides, can serve as substrates for CHIT1. The enzyme showed activity on LacdiNAc-TMR, an epimer of chitobiose, with a turnover of 0.4 s^−1^, which is comparable to the natural substrates pNP-chitotriose (0.5 s^−1^) and pNP-chitobiose (1.5 s^−1^). LacNAc-TMR, containing an N-acetyl group at the reducing end, was also a substrate but with a slower turnover (0.003 s^−1^). Interestingly, at higher concentrations of pNP-(GlcNAc)_2_, the 50 kDa recombinant enzyme displayed substrate inhibition, indicating reverse reactivity—known as transglycosylation. This is an important observation since, during inflammation, high concentrations of substrates for CHIT1 are present, accompanied by induced glycolysis, protein glycosylation, and glycan synthesis. Their study suggests that CHIT1 is capable of altering glycosylation associated with diseases, including asthma [34]. CHIT1, by acting on LacNAc-containing substrates, can potentially contribute to disease pathogenesis by altering the glycosylation pattern via either hydrolysis or transglycosylation on specific glycoside bonds [140,141]. Taking into consideration the fact that CHIT1 activity was shown to be elevated along with CCL18 in the model for allergic airway inflammation in asthma, these findings shed a new light on the role of chitinolytic enzymes in lung disease [142].

The relevance of the pathological role of CHIT1 in asthma was demonstrated by pre-clinical models with genetically modified mice lacking CHIT1 [143,144]. CHIT-1 is a marker of chronically activated macrophages, and recent data coming from unbiased omics experiments verified these findings. CHIT1 came up as one of the top hits in an atlas of pulmonary fibrosis created using single-cell RNA sequencing (scRNAseq) of fibrotic lungs from patients with ILDs and from healthy lungs. CHIT1 expression was restricted to a subset of macrophage clusters specific to fibrotic lungs and was not present in the healthy tissue [145]. In another single-cell sequencing and spatial transcriptomics study conducted on granulomas from patients with sarcoidosis, CHIT1 expression was found on proinflammatory macrophages in the center of granulomas [146]. These recent findings demonstrate the therapeutic potential of targeting CHIT1 in all disorders with chronic inflammation, including asthma as a perfect example [147,148,149].

## 4. Glycoenzyme Inhibitors Are Promising for the Treatment of Asthma

The potential of using specific glycoenzyme inhibitors as a promising therapeutic avenue for the management of asthma has not been fully explored, but it is justified by the presence of clinical-stage drugs targeting glycoenzymes in diseases like cancer and fibrosis [150,151]. In Table 1, we provide examples of the glycoenzymes implicated in the inflammatory pathways characteristic of asthma pathogenesis.

### 4.1. Inhibition of Chitotriosodase-1 (CHIT1)

There are a number of known chitinase inhibitors that have been developed in recent years, including numerous potent natural-product-derived inhibitors, such as the pseudosaccharide allosamidin and its derivatives [152], as well as cyclic peptides like argifin and argadin [153]. However, the practical application of these compounds for in vitro or in vivo studies was hindered by their high molecular weights, complex chemistry, and poor pharmacokinetic profiles. Nevertheless, small-molecule chitinase inhibitors, exemplified by Wyeth 1 or bisdionin C, do not have sufficient potency to show effects in vivo [154,155]. OATD-01, which was derived the hit compound Wayeth1, is the most potent and advanced inhibitor of CHIT1. It demonstrated pharmacological efficacy in a chronic HDM-induced lung inflammation model [41]. The anti-inflammatory effects of OATD-01 were manifested by the reduction in CD45 cells in bronchoalveolar lavage fluid (BALF) along with decreased chitinolytic activity in both the BALF and serum. Notably, OATD-01 mitigated airway remodeling by inhibiting the CHIT1-mediated active release of TGFβ1, subsequently reducing Th-2-dependent IL-13 production and subsequently decreasing collagen deposition in the extracellular space. Attenuated airway remodeling, which is characteristic of severe asthma, is explained by the cross-talk between CHIT1-producing macrophages and lung fibroblasts [41]. OATD-01 is currently in phase II clinical trials for lung sarcoidosis (NCT06205121).

### 4.2. Inhibition of Fucosyltransferases 2 and 8

Developing inhibitors for FUTs presents a promising strategy that has been verified so far in cancer models [156,157]. Clinical and preclinical progress is limited by the absence of crystal structures for certain FUTs, the complex transition state of FUT reactions, and the relatively low binding affinity for acceptor ligands [158]. 

### 4.3. Inhibition of Sialyltransferases (STs) ST6GAL1 and ST3GAL3

Currently, data come from studies using global sialylation inhibitors or inhibitors targeting other sialyltransferase isoforms in preclinical models of asthma or related airway diseases. Thus far, most inhibitors aim to modulate the overall sialylation levels rather than specifically targeting ST6GAL1 [66] or ST3GAL3 [69], which have potential applicability in asthma treatment.

### 4.4. Inhibition of Sialidase NEU1 and NEU3

Research focused on targeting sialidases is currently at the early stages of preclinical development [159,160,161,162]. In most instances, sialidase inhibitors have been suggested to imitate the transition state that arises during sialoside hydrolysis. The first transition-state analogue sialidase inhibitor, DANA (Neu5Ac2en), was designed to mimic the oxocarbenium ion-like transition state [163]. Analogues of DANA with specificity toward influenza viral neuraminidases display inhibitory activity and selectivity toward human isoforms NEU1–NEU4. C9-pentylamide analogues of DANA display moderate inhibitory activity against NEU1 in in vitro and in vivo models [164,165], and C5-hexanamido-C9-acetamido analog od DANA showed a highly improved potency with a Ki of 53 ± 5 nM and 330-fold selectivity [166]. C9-triazolyl DANA derivatives exhibited remarkably increased inhibitory activity for NEU3 [167].

### 4.5. Inhibition of Galectins

Glycosylation inhibitors may modulate the activity of glycan-binding proteins (lectins) involved in immune cell interactions and inflammatory signaling pathways, providing additional therapeutic targets for asthma treatment. Several compounds have been investigated as potential galectin inhibitors for the treatment of asthma and fibrotic lung diseases [168]. These inhibitors aim to modulate galectin binding activity and downstream inflammatory processes, potentially offering therapeutic benefits in asthma management. Several galectin-targeted therapies have entered clinical trials for indications, including cancer and cardiovascular and fibrotic diseases. The structures of inhibitors include carbohydrate mimetics, especially lactose-derived compounds described in detail in two recent reviews [169,170].

## 5. Conclusions

It is necessary to consider glycosylation during the development of therapies for asthma. Although we focused on the potential implications of CHIT1 inhibition in the context of asthma severity and airway remodeling, other enzymes hold therapeutic promise for glycosylation regulation, such as galectins, NEU1, and FUT2. Moreover, the broader implications of alterations in N-glycosylation patterns within specific cell subsets or protein groups are in the early stages of development and hold promise as potential markers for diagnosing asthma and other respiratory diseases [171]. Certain cell-specific proteins, whose dysregulation in glycosylation should be collectively monitored, could become future markers for the diagnosis, risk assessment, and severity verification of asthma [54,172]. Research on glycosylation alterations in asthma should help to stratify patients into distinct subsets for which we should be able to provide new personalized treatment.

## Figures and Tables

**Figure 1 biomolecules-14-00513-f001:**
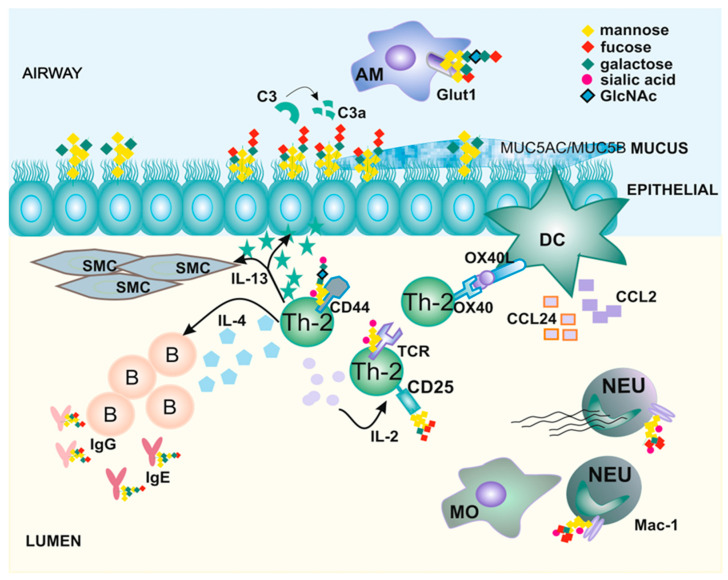
Changes in protein glycosylation shape the immune cells’ function and trafficking in asthma. The activation of Th-2 lymphocytes by dendritic cells (DC) leads to cytokine production and Th-2 cell expansion. IL-13 prompts the overgrowth of smooth muscle cells (SMCs), modifies functions of alveolar macrophages (MAs), and leads to the fucosylation of epithelial surface proteins. Due to the immune response, the hyperproduction of mucus occurs and affects the MUC5AC/MUC5B ratio. At the same time, IL-4 supports B cells in antibody production, which are also modified by glycans. Chemokine production increases the influx of myeloid cells, including monocytes (MOs) and neutrophils (NEUs). Modification by glycosylation receptors affects the cell’s trafficking (CD44 and Mac-1), function (TCR and CD25), and metabolism (Glut1).

**Table 1 biomolecules-14-00513-t001:** Key enzymes/lectins dysregulated in asthma and their glycoprotein substrates.

Enzyme/Lectin Class	Glycan-Modifying Enzyme	Catalyzed Reaction	Key Glycoprotein	Enzyme/Lectin Class
Fucosyltransferases (FUTs)	FUT2	transfers L-fucose from a GDP-fucose to the terminal galactose on both O- and N-linked glycans, i.e., residues on the mucin-type glycan chains	MUC5AC	Activation of mucin 5AC (MUC5AC) signaling pathway regulating the function of asthmatic airway smooth muscle cells (ASMCs) and participating in asthmatic airway remodeling [62,63]
	FUT8	catalyzes the transfer of fucose from GDP-fucose to the innermost GlcNAc in an α1-6 linkage	TGF-β1 and SPARC	Core fucosylation catalyzed by FUT8 is essential for TGF-β binding to TGF-β receptors [64]; loss of core-fucosylation of SPARC impairs collagen binding and contributes to COPD [65]
Sialyltransferases (STs)	ST6GAL1	catalyzes the transfer of sialic acid from the transfer of CMP-sialic acid to galactose-containing substrates	IL6, MUC4β, EGFR	Sialylation of Muc4beta n-glycans by St6gal1 orchestrates human airway epithelial cell differentiation associated with type-2 inflammation [66] St6gal1 and Alpha2–6 sialylation regulates IL-6 expression and secretion in chronic obstructive pulmonary disease [67] α2,6 sialylation promotes EGFR signaling by facilitating receptor oligomerization and recycling [68]
	ST3GAL3	transfers sialic acid from CMP-NeuAc in α-2,3 linkage, preferentially to Galβ1-3GlcNAc, but also to Galβ1-4GlcNAc and Galβ1-3GalNAc termini on glycoproteins and glycolipids	MUC5B	Endogenous airway mucins carry glycans that bind Siglec-F and induce eosinophil apoptosis [69]
Sialidases (Sas)	NEU1	hydrolyzes α-(2→3)-, α-(2→6), and α-(2→8)-glycosidic linkages of terminal sialic residues	CD44 TLR1-4, ICAM-1, SCD15 FCRy	A crucial role of sialidase Neu1 in hyaluronan receptor function of CD44 in T helper type 2-mediated airway inflammation of murine acute asthmatic model [70,71]; NEU1 participates in regulation of cell signaling by desialylating plasma membrane receptors [72]
Lectins	Galectin3	the specific binding of β-galactosides; cross-linking of N-acetyllactosamine (LacNAc) of cell surface receptors	IgE	New regulatory roles of galectin-3 in high-affinity IgE receptor signaling [73]
	Galectin9		IgE	Galectin-9 is a high affinity IgE-binding lectin with anti-allergic effect by blocking IgE-antigen complex formation [74]

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
