# Peer review of "The Exploitation of the Glycosylation Pattern in Asthma: How We Alter Ancestral Pathways to Develop New Treatments"

_biomolecules, 2024, doi:10.3390/biom14050513_

Round 1

Reviewer 1 Report

Comments and Suggestions for Authors

Peer review of the document entitled "Exploitation of glycosylation pattern in asthma: how we alter ancestral pathways to develop new treatments", by Angelika Muchowicz, Agnieszka Bartoszewicz and Zbigniew Zaslona.

General comments: 

The aim of the above-mentioned review was to summarize the current available evidence of protein glycosylation in asthma to develop potential novel biomarkers and specific therapy. The manuscript lays within the journal's scope.

Specific Comments:

Abstract: Despite informative the abstract should concluded in a different manner (not so openly finished).

Main text:

Line 44: The sentence "This old-fashioned way..." is not appropriate and needs to be rephrased.

Line 54:  The whole sentence "While protein glycosylation..." should be clarified and rephrased as a more elegant description.

Line 167: The description of "a type 2-ultra-high endotype" (if included) should be  quantitatively defined in  terms of biomarkers and compared to the type 2-high endotype.   

Line 175: The sentence "Although severa asthma is not precisely..." needs to be clarified and rephrased.

Line 390: Conclusions

This section should be further expanded. A summarized comment on glycosylation and the  potential future perspective on both asthma diagnosis and therapeutics may be also added. 

Comments on the Quality of English Language

Minor editing of English language required

Author Response

Dear Reviewer,

Thank you for your time to improve our manuscript, please find attached reply to your comments which we addressed in a point-by-point responses and submitted a revised version of the manuscript according to your suggstions.

On behalf of the co-authors,

Zbigniew Zaslona 

Reviewer 2 Report

Comments and Suggestions for Authors

Reasearch gap and reason behind this review could be more elaborative.

Improvement of the title is suggested and type of review and search engines for literature search should be mentioned.

194 line heading sequence should be rechecked.

Comments on the Quality of English Language

Writing language can be more simplified.

Author Response

(The authors gave the same response as above.)

Reviewer 3 Report

Comments and Suggestions for Authors

The manuscript “Exploitation of glycosylation pattern in asthma: how we alter ancestral pathways to develop new treatments” deals with the reviewing literature devoted to the glycosylation in asthma.

The topic discussed is interesting for the developing new approaches for treatment patients with asthma.

I would like to make a few comments:

1.         Please state the aim of the study in the introduction.

2.         lines 60-62:

 Protein glycosylation … affecting over 50% of known proteins.

Comment: A link to a source confirming the data provided is required.

3.         lines 75-76:

The majority of the process of protein glycosylation occurs in the endoplasmic reticulum and Golgi apparatus, as well as the cytoplasm and nucleus.

Comment: No glycosylation occurs in the nucleus

4.         lines 97-98:

The building blocks for glycan synthesis are activated nucleotide sugars, which are synthesized in the cytoplasm or nucleus of the cell.

Comment:

This is wrong. Nucleotide sugars are not synthesized in the nucleus:

Lane, A. N., & Fan, T. W. (2015). Regulation of mammalian nucleotide metabolism and biosynthesis. Nucleic acids research, 43(4), 2466–2485.  

Zhao H., French J., Fang Y., Benkovic S.J. The purinosome, a multi-protein complex involved in the de novo biosynthesis of purines in humans. Chem. Commun. 2013;49:4444–4452.

5.         Information on the impact of genetic backgrounds on the pathogenesis and treatment of asthma is incomplete. It is advisable to discuss SNPs and mutations of the enzymes involved in glycosylation observed in asthma.

More than 100 gene polymorphisms are associated with asthma according latest research.  It is necessary to analyze whether these 100 genes include genes for glycosylation enzymes:

Shi F, Zhang Y, Qiu C. Gene polymorphisms in asthma: a narrative review. Ann Transl Med. 2022 Jun;10(12):711. doi: 10.21037/atm-22-2170.

6.         Discuss please, are there any glycans that can be detected as markers in asthma

Discuss the article:

Yeh YL, Wu WC, Kannagi R, Chiang BL, Liu FT, Lee YL. Sialyl Glycan Expression on T Cell Subsets in Asthma: a correlation with disease severity and blood parameters. Sci Rep. 2019 Jun 20;9(1):8947. doi: 10.1038/s41598-019-45040-2.

7.         Discuss the usage of specific glycoenzyme inhibitors as promising approach for the asthma treatment.

8.         When you are writing about IgE, it is necessary to mention that the binding of serum IgA to FcαRI can  inhibit the IgE-mediated asthma. Moreover  severe asthma patients present lower serum IgA levels compared to healthy controls:

Ding L, Chen X, Cheng H, Zhang T and Li Z (2022) Advances in IgA glycosylation and its correlation with diseases. Front. Chem. 10:974854. doi: 10.3389/fchem.2022.974854

Sánchez Montalvo A, Gohy S, Rombaux P, Pilette C and Hox V (2022) The Role of IgA in Chronic Upper Airway Disease: Friend or Foe? Front. Allergy 3:852546. doi: 10.3389/falgy.2022.852546

Author Response

(The authors gave the same response as above.)
